# How do soundboard-trained dogs respond to human button presses? An investigation into word comprehension

Amalia P. M. Bastos[1,2], Ashley Evenson[3,4], Patrick M. Wood[1,2,4], Zachary N. Houghton[5], Lucas Naranjo[4,6], Gabriella E. Smith[7,8], Alexandria Cairo-Evans[1], Lisa Korpos[1,4], Jack Terwilliger[1], Sarita Raghunath[1], Cassandra Paul[1], Hairou Hou[1], Federico Rossano[1] *

1 Department of Cognitive Science, University of California, San Diego, San Diego, California, United States of America, 2 Department of Psychological & Brain Sciences, Johns Hopkins University, Baltimore, MD, United States of America, 3 College of Arts and Sciences, Canisius College, Buffalo, New York, United States of America, 4 FluentPet, Inc, San Diego, California, United States of America, 5 Department of Linguistics, University of California, Davis, Davis, California, United States of America, 6 Statistics and Operational Research Department, Universitat de València, Valencia, Spain, 7 Comparative Cognition, Messerli Research Institute, University of Veterinary Medicine Vienna, Medical University of Vienna, University of Vienna, Vienna, Austria, 8 School of Psychology and Neuroscience, University of St. Andrews, St. Andrews, United Kingdom

* frossano@ucsd.edu

**Data Availability Statement:** Study pre-registration is deposited in the Open Science Framework, https://osf.io/tskcq Data and analysis

## Abstract

Past research on interspecies communication has shown that animals can be trained to use Augmentative Interspecies Communication (AIC) devices, such as soundboards, to make simple requests of their caretakers. The recent uptake in AIC devices by hundreds of pet owners around the world offers a novel opportunity to investigate whether AIC is possible with owner-trained family dogs. To answer this question, we carried out two studies to test pet dogs' ability to recognise and respond appropriately to food-related, play-related, and outside-related words on their soundboards. One study was conducted by researchers, and the other by citizen scientists who followed the same procedure. Further, we investigated whether these behaviours depended on the identity of the person presenting the word (unfamiliar person or dog's owner) and the mode of its presentation (spoken or produced by a pressed button). We find that dogs produced contextually appropriate behaviours for both play-related and outside-related words regardless of the identity of the person producing them and the mode in which they were produced. Therefore, pet dogs can be successfully taught by their owners to associate words recorded onto soundboard buttons to their outcomes in the real world, and they respond appropriately to these words even when they are presented in the absence of any other cues, such as the owner's body language.

## Introduction

The use of Augmentative Interspecies Communication (AIC) devices has gained popularity among pet owners in recent years, especially among dog owners [1–3]. Most of these AIC

scripts are available from Github, https://github.com/znhoughton/comprehension.

**Funding:** A.P.M.B. is supported by the Johns Hopkins Provost's Postdoctoral Fellowship Program.

**Competing interests:** A.P.M.B., P.M.W., Z.N.H., J. T., G.E.S., & L.K. have previously consulted for FluentPet, Inc., a company that produces AIC devices for pets. A.E. & L.N. are employees of FluentPet, Inc.

devices consist of soundboards with buttons that, when pressed, produce pre-recorded words in the owner's voice. Most owners report that they train their dogs using a technique that has become known as modelling: owners demonstrate the actions associated with each button repeatedly until their animals make presses themselves, at which point the same actions are performed by the owners, regardless of whether the animal's press was intentional or accidental [2]. Over time, animals may press the buttons on their soundboards more frequently, which may in turn provide them with more control over their daily lives and environments.

Although interspecies communication through soundboard-like AIC devices has been adopted to varying degrees of success with apes, dolphins, and other species (for a review, [1]), this has primarily been undertaken by researchers and professional animal trainers. In the scientific literature, only a single dog named Sofia has been shown to use buttons to request actions such as going on a walk or playing, following training by a professional dog trainer [4, 5]. The training methods used with Sofia are considerably different from the modelling approach that is most common among pet owners: Sofia was first taught to tap her AIC device with her paw on command, which was eventually phased out once she began doing so spontaneously [4]. Given the nature of Sofia's training, it is also possible that Sofia experienced less cueing from her trainer than other owner-trained pet dogs, whose performance may be more susceptible to the Clever Hans effect [6, 7].

Crucially, communicative production of words, such as that demonstrated by Sofia, must be preceded by the comprehension of said words. Although Sofia is the only dog ever shown to produce requests through an AIC device, many more dogs are capable of responding appropriately to spoken human signals. Most pet dogs are trained to respond to at least a few vocal signals from their owners such as "sit" and "lie down" [8, 9], but some dogs such as Rico and Chaser can learn tens or hundreds of names for individual objects ([10–17]; although learning object names may be more challenging to dogs than learning words associated with actions, see [18]). Given that owner-trained soundboard-using dogs have not yet been tested in controlled experimental contexts, we currently do not have any evidence to suggest that they have successfully associated the words produced by their soundboards to their respective consequences, let alone that they produce said words communicatively.

Another potential issue surrounding owner-trained soundboard-using dogs concerns the reliability of citizen science data [19]. Although thousands of pet owners currently contribute regular data on their animals' soundboard use to scientific research [2], the extent to which reported interactions are cued, inaccurately reported, or cherry picked by owners is unclear and a matter under current investigation. Consequently, testing soundboard-trained dogs under controlled experimental conditions is a necessary step towards assessing this form of interspecies communication.

Given the novelty of this approach, the nature of most of the data on this form of interspecies communication, and our limited knowledge of dogs' capacity to acquire associations in this context, in the present study we pursue a thorough and multifaceted investigation into the comprehension of button presses by humans. We begin from the premise that owner-trained soundboard-using dogs are probably more likely to associatively learn the words for routine activities or events (such as learning that the word "outside", when spoken or pressed by a human, is usually followed by a door being opened to the backyard) than some of the more abstract words sometimes provided on owner-trained dogs' soundboards (such as "tomorrow" or "later"). Given that dogs' soundboards are determined by their owners and therefore are composed of buttons with different words, and have different layouts (for a detailed explanation, see [2]), here we select three of the most commonly occurring routine-related word concepts across all participants (namely, food-related, play-related, and outside-related words) and choose the most common variants within these concepts (OUT or OUTSIDE for outside-

related words, PLAY or TOY for play-related words, and FOOD or EAT or DINNER or HUN-GRY for food-related words) for our experiments (as pre-registered in the Open Science Framework: https://osf.io/tskcq). Over the course of two complementary experiments, we use these three word concepts, as well as a nonsense word (the nonce word "DAXING") to answer five fundamental questions about owner-trained soundboard-using dogs' comprehension of the words on their soundboards.

First, using both a researcher-led in-person experiment and a remotely conducted citizen science experiment, we test whether dogs can recognise and respond appropriately to the words recorded onto their soundboards. We hypothesise that, if dogs have associatively learnt the connection between these words and the outcomes they usually entail, then they should behave in anticipation of the actions or events indicated by these words. For example, upon hearing the word "PLAY", dogs should be more likely to walk to their toy box and pick up a toy, than if they had heard "FOOD". Conversely, we should see fewer looks to a food bowl following the word "PLAY" than following "FOOD". Given that these words might typically be produced by owners concurrently with many other contextual cues (for example, "FOOD" might be produced most commonly at specific mealtimes, while the owner fills up the dog's bowl), in our experimental trials we strip all words of any additional contextual cues, to ensure that dogs' responses occur specifically in response to the words themselves.

Second, we use the two experiments to compare dogs' responses to a word when it is produced by the owner and by an unfamiliar person (a researcher). We hypothesise that, if dogs expect words to precede their accompanying actions, and can do so without any additional contextual or owner-provided cues, then they should respond similarly regardless of the person producing the word. Given that words on dogs' buttons usually consist of recordings of their owner's speech (as was the case for every dog participating in this study), and that differences in the acoustic properties of different people's speech can conceivably generate differences in dogs' behavioural responses unrelated to their comprehension of words, we compare their responses to button presses by the owner and presses of those same buttons by an unfamiliar person.

Third, a comparison between the two experiments–particularly in terms of dogs' responses to button presses by their owners and an unfamiliar person–can also inform whether the results of citizen science studies on soundboard-trained dogs are broadly comparable to in-person researcher-led studies, and therefore establish whether future work with this population could be carried out through data collected remotely by citizen scientists.

Fourth, we also compare dogs' responses to owners' button presses and owners' vocal production of these same words. If dogs respond appropriately and equivalently to words regardless of their mode of production (whether they are spoken directly by the owner or produced by a button press), then this would suggest that dogs are associating words' consequences (e.g., the actions following words, such as a play interaction following "PLAY") to the words (speech or speech recordings) themselves, rather than attending primarily to other cues such as the location of the buttons (which would predict contextually-appropriate responses when buttons are pressed, but not when the same words are spoken out loud). On the other hand, if dogs respond appropriately to owners' button presses of words but not to their vocal production, then it is likely that dogs associate button locations–or some other property of the buttons–to their accompanying actions or events, disregarding button audio.

Finally, because of recent research suggesting that dogs tilt their heads in response to familiar words more so than unfamiliar ones [20], and that this is a lateralized behaviour, we investigated subjects' head tilts throughout the two experiments. We hypothesise that, if dogs do in fact tilt their heads in response to words that they recognise, and soundboard-trained dogs recognise the words recorded onto their buttons, then we should see more head-tilting behaviour

in response to the known words (i.e., outside-related, play-related, and food-related words) than to a nonce word ("DAXING"). This result should hold regardless of whether words are produced by button presses or spoken by their owner. If this is true, then this study can also provide a larger sample size to inform whether this head-tilting behaviour is more commonly left-lateralized or right-lateralized, or, as in the original study, this is split evenly across subjects. If, on the other hand, we find evidence to suggest that soundboard-trained dogs do respond appropriately to known words but do not tilt their heads upon hearing them, this might instead suggest that there are population or breed differences between soundboard-trained dogs and the "gifted world learner" [15] dogs in the study conducted by Sommese and colleagues [20].

In sum, the present study aims to investigate five principal questions: (1) whether dogs can recognise and respond appropriately to the words recorded on their soundboards; (2) whether they exhibit these responses even in the absence of other contextual or owner-produced cues, even when they are produced by an unfamiliar person; (3) whether citizen science studies and in-person studies can produce comparable results in this population of subjects; (4) whether dogs attend to speech specifically in this context, or other cues; and (5) whether owner-trained soundboard-using dogs tilt their heads in response to familiar words in the same way that dogs trained to recognise object names do.

## Methods

### Experiment 1: In person study

**Subjects.** For the in-person study (IPS), subjects were 30 dogs (14 males, age: M = 3.52 years, SD = 2.91; see Table 1 for subject information). No subjects were excluded from analyses. All dogs were family pets trained by their owners to use soundboard AIC devices, and whose soundboards contained the words OUT/OUTSIDE, PLAY/TOY, and FOOD/EAT/ DINNER/HUNGRY. Owners were asked to habituate their dogs to people wearing sunglasses, hats, and other head accessories prior to the study date, in their own time, given that the study would involve owners wearing multiple accessories on their faces. Dogs were tested in their home settings with their owners present; owners were not told the purpose of the study until testing was concluded. All subjects lived primarily indoors and were fed indoors and the study received ethics approval from the UCSD IACUC (protocol no. S21098).

**Procedure.** The study was conducted in dogs' homes, in the room where their soundboard was typically located. Upon arrival, one researcher (Experimenter 1, hereafter E1) waited outside the house whilst the other (Experimenter 2, hereafter E2) went into the home and greeted the owner and their dog, before placing the dog either in the backyard or in a room in the house other than the one where the soundboard was located.

Once the dog was out of sight, E2 placed three large coloured stickers over the three buttons containing one of each of the words of interest: (1) "OUT" or "OUTSIDE"; (2) "PLAY" or "TOY"; and (3) "FOOD" or "EAT" or "DINNER" or "HUNGRY". This ensured that, at test, E1 could not tell the buttons' identity based on any writing or symbols visible on the buttons. E2 then asked that the owner record their voice onto a new button (that matched the size and style of the dog's existing soundboard buttons) for the word "DAXING" and placed that on the soundboard, with another coloured sticker. Sticker colours were randomly assigned to buttons between subjects, such that E1 would not have known the identity of any buttons based on the colour of the stickers placed on them. Stickers were made from green, red, yellow, and blue cardstock paper and adhesive putty. Owners were shown a novel action–placing their hands on their head and spinning in a slow circle–which they were told they would have to perform at some point later in the study.

**Table 1. List of subjects that participated in Experiment 1.** Age at time of testing is provided in years, rounded to the nearest decimal point, as per owner reports. For mixed breed dogs, up to three primary component breeds (based on owner reports or genetic testing) are provided in brackets.

| Subject | Breed | Sex | Age (Years) | Eligible Buttons | Testing Location |
|---------|-------|-----|-------------|------------------|------------------|
| Adora | Border Collie | F | 3.0 | OUTSIDE, PLAY, EAT | Living Room |
| Alvin | Mixed Breed (*Golden Retriever, Pug, Beagle*) | M | 9.6 | OUTSIDE, PLAY, FOOD | Living Room |
| Bastian | Mixed Breed (*Rat Terrier, Pomeranian, Miniature Poodle*) | M | 4.1 | OUTSIDE, PLAY, HUNGRY | Living Room |
| Buster | Mixed Breed (*French Bulldog, Boston Terrier*) | M | 5.8 | OUTSIDE, PLAY, EAT | Living Room |
| Charlie | Yorkshire Terrier | M | 3.2 | OUTSIDE, PLAY TOY, FOOD | Living Room |
| Chester | Mixed Breed (*Labrador Retriever, Boston Terrier, Pomeranian*) | M | 14.0 | OUTSIDE, PLAY, FOOD | Living Room |
| Chuckles | Mixed Breed (*American Cocker Spaniel, Miniature Poodle*) | M | 2.9 | OUTSIDE, PLAY, FOOD | Living Room |
| Dillon | Mixed Breed (*Labrador Retriever, German Shepherd Dog, Chow Chow*) | M | 2.8 | OUTSIDE, PLAY, FOOD | Living Room |
| Eleanor | German Shepherd Dog | F | 7.5 | OUTSIDE, PLAY, HUNGRY | Living Room |
| Eleanor | Mixed Breed (*Bernese Mountain Dog, Standard Poodle*) | F | 1.8 | OUTSIDE, PLAY, FOOD | Living Room |
| Frank | Mixed Breed (*Affenpinscher, Miniature Schnauzer*) | M | 2.8 | OUTSIDE, PLAY, FOOD | Living Room |
| Gibson | Mixed Breed (*Bernese Mountain Dog, Miniature Poodle*) | M | 1.3 | OUTSIDE, PLAY, HUNGRY | Living Room |
| Gracie | Mixed Breed (*Miniature Poodle, Labrador Retriever*) | F | 3.2 | OUTSIDE, PLAY, HUNGRY | Living Room |
| Izzy | Miniature American Shepherd | F | 2.7 | OUTSIDE, PLAY, HUNGRY | Living Room |
| Jumbo | Miniature American Shepherd | F | 2.2 | OUTSIDE, PLAY, HUNGRY | Entrance Hallway |
| Juniper | Mixed Breed (*Miniature Poodle, Chihuahua, other*) | F | 8.5 | OUTSIDE, PLAY, FOOD | Home Office |
| Juno | Mixed Breed (*Pug, American Bully, American Pit Bull Terrier*) | F | 1.3 | OUTSIDE, PLAY, EAT | Living Room |
| Kitsune | Mixed Breed (*Bernese Mountain Dog, Miniature Poodle*) | F | 0.6 | OUTSIDE, PLAY, HUNGRY | Living Room |
| Lobo | Alaskan Klee Kai | M | 5.2 | OUTSIDE, PLAY, HUNGRY | Living Room |
| Luna | Miniature American Shepherd | F | 2.7 | OUTSIDE, PLAY, HUNGRY | Living Room |
| Misty | Mixed Breed (*Siberian Husky, Australian Cattle Dog*) | F | 0.6 | OUTSIDE, PLAY, EAT | Living Room |
| Mokai | Border Collie | M | 2.5 | OUTSIDE, PLAY, EAT | Dining Room |
| Oski | Pug | M | 3.5 | OUTSIDE, PLAY, FOOD | Living Room |
| Parker | Mixed Breed (*Beagle, American Eskimo Dog, Chihuahua*) | F | 1.5 | OUTSIDE, PLAY, HUNGRY | Living Room |
| Pepita | Mixed Breed (*American Foxhound, Australian Cattle Dog, Beagle*) | F | 1.9 | OUTSIDE, PLAY, FOOD | Dining Room |
| PJ | Mixed Breed (*Standard Poodle, Golden Retriever*) | M | 1.3 | OUTSIDE, PLAY, EAT | Living Room |
| Rey | Mixed Breed (*Beagle, Dachshund, Miniature Pinscher*) | F | 3.5 | OUTSIDE, PLAY, EAT | Living Room |
| Scout | Mixed Breed (*German Shepherd Dog, Boxer, American Pit Bull Terrier*) | F | 2.7 | OUTSIDE, PLAY, HUNGRY | Living Room |
| Simon | Standard Poodle | M | 0.6 | OUTSIDE, PLAY, FOOD | Living Room |
| Zero | Shiba Inu | F | 2.3 | OUTSIDE, PLAY, HUNGRY | Living Room |

The dog was then brought back into the room where their soundboard was located, and E1 was invited into the home and allowed to greet the dog and owner. E2 then set up the camera recording equipment (Panasonic Full HD Camcorder HC-V180K, GoPro Hero7) in the room, and an Anmeate Baby Monitor facing the soundboard, such that multiple camera angles of the room were recorded (Fig 1).

While E1 explained the experimental procedure to the owner and ensured the dog was comfortable and behaving normally in their presence, E2 identified a different room in the house to hide so they could remotely monitor the study (through a live feed of the baby monitor). Next, the owner was asked to wear a sleep mask and noise-cancelling headphones and ensure that the dog was not fearful of either accessory. Finally, E2 handed E1 their headphones and set both headphone volumes such that that neither E1 nor the dog's owner could hear E2 speaking loudly from a short distance, therefore ensuring that they would also be unable to hear the recordings on any of the buttons when they were pressed during a trial.

Trials began with E1 and the owner sitting in the soundboard room whilst the dog ranged freely around the room. The owner wore a sleep mask and listened to music on noise-

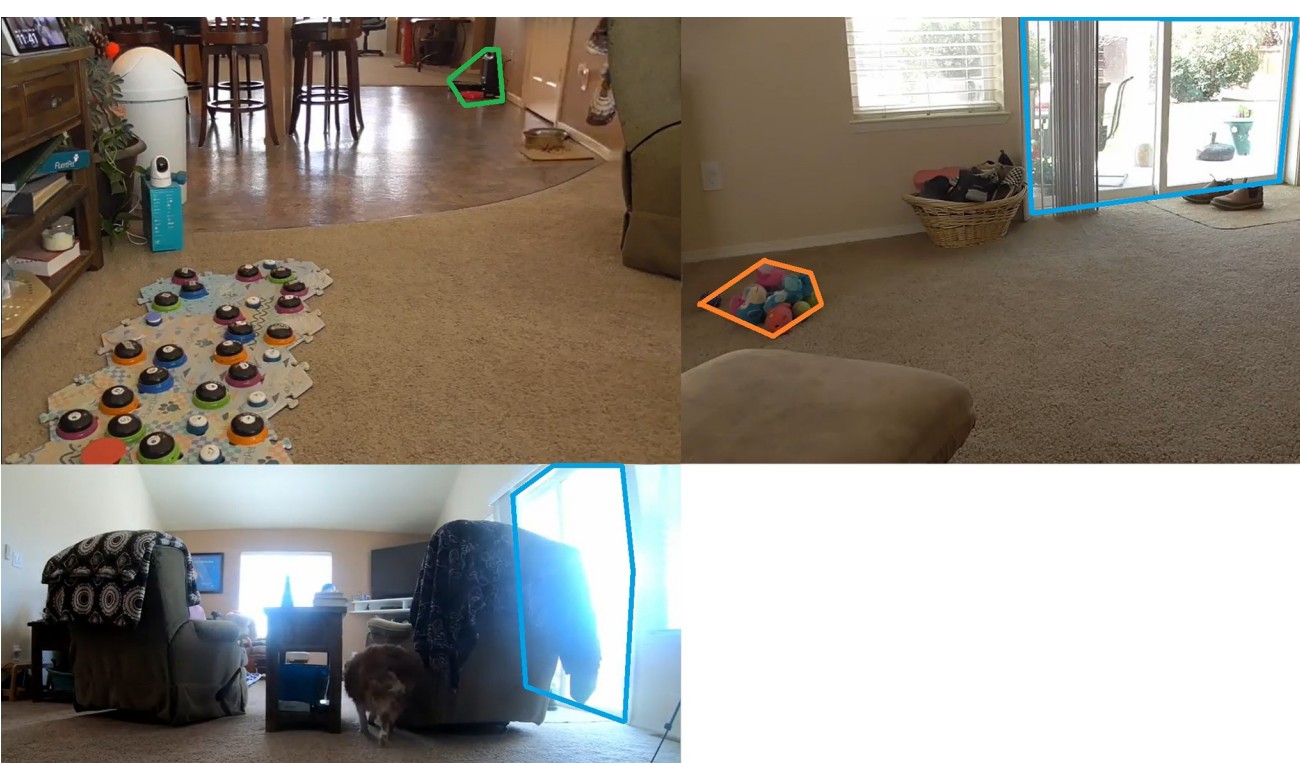

**Fig 1. Set up of a trial for the in-person study.** The door used for the button concept "OUTSIDE" is outlined in blue, the dog's toys are outlined in orange, and the food bowl is outlined in green. During trials for this subject, for example, the owner sat in the reclining chair on the bottom left of the frame, facing away from the soundboard, while wearing noise-cancelling headphones, a face mask (as a Covid-19 precaution), and a black sleep mask.

cancelling headphones so they could not see any of the trial procedures. This ensured that owners were blind to the study's hypotheses and predictions but were still present in the room to ensure that dogs were comfortable and behaving normally.

At the start of each trial, E2 remotely triggered a sound file to E1's headphones stating the colour of the sticker for the button they would press in that trial, then remotely started playing music into the headphones of both E1 and the dog's owner. E1 looked straight ahead until they heard a beep in their headphones, again triggered via Bluetooth by E2. At this point, E1 stood up and began the trial: E1 walked to the same place by the soundboard each time, pressed the button covered by the sticker colour they were given, stepped away from the button again, and looked in the direction of the dog. E1 froze in this position for 1 minute while they waited for the next beep in their headphones, at which point they turned around to face the nearest wall and closed their eyes. This final beep and turn concluded the trial.

Following the end of the trial, E2 triggered an audio file in the owner's headphones stating that they could remove their headphones and sleep mask and gave the owner instructions on an action to perform. Unbeknownst to the owner, this action always matched the button recently pressed by E1. For "OUT" or "OUTSIDE", the owner was asked to let their dog outside as they normally would (e.g., open the door to the backyard so the dog could go out). For "PLAY" or "TOY", the owner was asked to briefly engage their dog in play with a toy. For

"FOOD", "EAT", "DINNER", or "HUNGRY", the owner was asked to place a small amount of their dog's usual food in their food bowl. For "DAXING", the owner performed the novel action they practiced earlier, namely placing both hands on the top of their head and spinning in a slow circle. While the owner performed their action, E1 remained with their eyes closed, listening to music through noise-cancelling headphones, so that they would not be unblinded to the button they just pressed. Once the owner performed their action, E2 came out of their hiding place and asked the owner to return the room to its starting state (e.g., closing any doors that were opened, placing any toys back where they were at the start of the trial), and returning to their starting position in the room. E2 then walked over to E1 and tapped them on the shoulder, so that E1 knew they could remove their headphones, open their eyes, and turn back around.

## Experiment 2: Citizen science study

**Subjects.** In the citizen science study (CSS), subjects were a new group of 29 dogs (9 males, age: M = 2.95 years, SD = 1.91; see Table 2 for subject information). Another 3 subjects (3 males) completed the study but had their data excluded due to procedural errors, and a

**Table 2. List of subjects that participated in Experiment 2.** Age at time of testing is provided in years, rounded to the nearest decimal point, as per owner reports. For mixed breed dogs, up to three primary component breeds (based on owner reports or genetic testing) are provided in brackets.

| Subject | Breed | Sex | Age (Years) | Eligible Buttons |
|---------|-------|-----|-------------|------------------|
| BeBe Ivy | Mixed Breed (*American Foxhound, Boxer, American Pit Bull Terrier*) | F | 3.3 | OUTSIDE, PLAY, HUNGRY |
| Birdy | Mixed Breed (*American Pit Bull Terrier, other*) | F | 1.8 | OUTSIDE, PLAY, EAT |
| Dexadrine | German Shepherd Dog | F | 3.0 | OUTSIDE, PLAY, EAT |
| Dio | Miniature American Shepherd | F | 2.2 | OUTSIDE, PLAY, FOOD |
| Freya | Australian Shepherd | F | 2.5 | OUTSIDE, PLAY, HUNGRY |
| Ila | Mixed Breed (*Standard Poodle, Golden Retriever*) | F | 3.7 | OUTSIDE, PLAY, HUNGRY |
| Juno | Mixed Breed (*Springer Spaniel, Labrador Retriever*) | F | 2.2 | OUTSIDE, PLAY, DINNER |
| Juno | Mixed Breed (*American Pit Bull Terrier, Labrador Retriever, Australian Shepherd*) | F | 1.1 | OUTSIDE, PLAY, EAT |
| Kida | Australian Shepherd | F | 4.7 | OUTSIDE, PLAY, FOOD |
| Kiki | Border Collie | F | 2.1 | OUTSIDE, PLAY, FOOD |
| Kipo | French Bulldog | M | 1.3 | OUTSIDE, PLAY, FOOD |
| Lina | Mixed Breed (*Australian Kelpie, Pembroke Welsh Corgi*) | F | 1.6 | OUTSIDE, PLAY, FOOD |
| Lola | Mixed Breed (*Border Collie, Labrador Retriever, Standard Poodle*) | F | 2.2 | OUTSIDE, PLAY, FOOD |
| Lucy | Havanese | F | 1.7 | OUTSIDE, PLAY, FOOD |
| Lulu | Mixed Breed (*American Pit Bull Terrier, Labrador Retriever, Australian Shepherd*) | F | 1.8 | OUTSIDE, PLAY, FOOD |
| Monty | Staffordshire Bull Terrier | M | 0.8 | OUTSIDE, PLAY, EAT |
| Moxie | German Shepherd Dog | F | 2.2 | OUTSIDE, PLAY, FOOD |
| Neko | Siberian Husky | M | 4.9 | OUTSIDE, PLAY, FOOD |
| Nita | German Shepherd Dog | F | 5.2 | OUTSIDE, PLAY, FOOD |
| Oliver | Golden Retriever | M | 2.4 | OUTSIDE, PLAY, FOOD |
| Pepper | Mixed Breed (*Maltese, Miniature Poodle*) | F | 1.9 | OUTSIDE, PLAY, FOOD |
| Pinot | Mixed Breed (*Australian Shepherd, Labrador Retriever, Standard Poodle*) | M | 2.1 | OUTSIDE, PLAY, EAT |
| Rhea Rose | Standard Poodle | F | 1.0 | OUTSIDE, PLAY, FOOD |
| Roscoe | Mixed Breed (*Staffordshire Bull Terrier, Cavalier King Charles Spaniel*) | M | 5.0 | OUTSIDE, PLAY, FOOD |
| Sadie | Mixed (*German Shepherd Dog, Staffordshire Bull Terrier, Chow Chow*) | F | 4.9 | OUTSIDE, PLAY, EAT |
| Samwise | Labrador Retriever | M | 1.7 | OUTSIDE, PLAY, EAT |
| Waffle | Mixed Breed (*Chihuahua, Miniature Poodle, Pug*) | M | 2.0 | OUTSIDE, PLAY, HUNGRY |
| Zodiak | Mixed Breed (*American Pit Bull Terrier, Staffordshire Bull Terrier, Chow Chow*) | M | 9.3 | OUTSIDE, PLAY, FOOD |
| Zoe | German Shepherd Dog | F | 7.0 | OUTSIDE, PLAY, FOOD |

further 8 subjects (4 males) were recruited for the study but never completed it. All dogs were owner-trained family pets with soundboards containing the same words as required in Experiment 1. Owners were asked to habituate their dogs to sunglasses prior to beginning the study, since owners were asked to wear mirrored sunglasses throughout all trials. Dogs were tested in their home settings by their owners, who were not told the purpose of the study until testing was concluded. As before, all dogs lived primarily indoors and were fed their meals indoors, and the study received ethics approval from the UCSD IACUC (protocol no. S21098).

**Procedure.** Owners were given study instructions in three formats: an explanatory video demonstrating study procedures; written materials with step-by-step instructions; and a 15-minute zoom call with a researcher who talked them through the study instructions and answered any remaining questions. As with the in-person study, owners were not told the purpose of the experiment until after they concluded their participation and submitted video data of all trials to the research team. Debriefs were also done by an experimenter through another 15-minute zoom call. Given that they administered the experimental trials to their own dogs, owners could not be made blind to conditions.

In each trial, owners either pressed the button for, or spoke, one of the words used in Experiment 1, namely: (1) "OUT" or "OUTSIDE"; (2) "PLAY" or "TOY"; and (3) "FOOD" or "EAT" or "DINNER" or "HUNGRY", and (4) the nonce word "DAXING". Each word was produced twice by the owner over the course of the study, once spoken and once pressed. Owners carried out two trials per day, allowing an interval of at least 30 minutes between the two trials within a day, over the course of four days, which were spread over a maximum of 2 weeks. Therefore, dogs in this study experienced a total of eight conditions: a spoken outside word, a pressed outside word, a spoken play word, a pressed play word, a spoken food word, a pressed food word, the word "DAXING" produced by a button press, and "DAXING" spoken by their owner. Trial orders were randomised across subjects.

Before the start of each trial, owners were asked to move their dog to a different room of the house where they could not see their soundboard. Owners then recorded a new button for the word "DAXING" and placed it in an empty slot on their dog's soundboard. The dog was brought back into the room where their soundboard was located for the start of the trial.

Each trial began with the owner standing in the same location next to the dog's soundboard. They then either pressed the button or spoke the word specified for that trial. Owners then refrained from interacting with their dog for the next minute; owners were specifically instructed not to speak to, point to, react to, gesture at, or otherwise interact with their dog or their environment for a full minute. At the end of each trial, owners were asked to perform the matching action for the pressed or spoken word, as specified by their participant sheet (i.e., as in Experiment 1, owners were asked to open the door and let their dog into the backyard following the production of the word "OUTSIDE", and spin slowly with their hands on their heads following the word "DAXING"). Following their post-trial action, owners again took their dogs out of the room and returned the soundboard to its usual configuration, removing the button for the word "DAXING", such that dogs could not press the "DAXING" button in the intervals between trials and on non-trial days. The full trial time was recorded on video by the owners, sometimes from multiple camera angles.

**Coding and analyses.** Videos for both studies (Experiment 1 and Experiment 2) were processed prior to blind coding to ensure that blind coders could never observe the button press produced by E1 (Experiment 1) or the button press or word spoken by the owner (Experiment 2), as well as the post-trial action performed by the owner. Video files were renamed so as to remove any trial information. Videos from different camera angles of the same trial were time-matched and aligned side-by-side into a single video file. E1 and the owner were occluded by a black rectangle such that blind coders could not see any humans on the screen, and the video

**Table 3. Ethogram of behaviours coded for both experiments.** We predicted that dogs would show more outside-directed behaviours in response to "OUT" or "OUTSIDE" words, more toy-directed behaviours in response to "PLAY" or "TOY" words, and more food-directed behaviours in response to "FOOD", "EAT", "DINNER", or "HUNGRY" words.

| Undirected Behaviours | | |
|---|---|---|
| **Behaviour** | **Modifiers** | **Behaviour Description** |
| Lip/Nose Licking | N/A | Contact between tongue and dog's nose or lips (including tongue flicks). |
| Head Tilting | Left, Right | Dog momentarily holds its head diagonally rather than vertically, to either side. |
| Door-Directed Behaviours | | |
| **Behaviour** | **Behaviour Description** | |
| Moving Towards Door | Distance between dog and door decreases, as dog moves with their head oriented towards the door. It should be marked from the moment the dog lifts their paw to make the movement towards the door. | |
| Looking at Door | Dog faces the door and looks at it for >1 second. | |
| Touching Door | Dog intentionally physically interacts with the door: namely by scratching at, pawing at, or nudging it. | |
| Sitting/Lying by Door | Dog changes their position to sitting by or laying down by the door. | |
| Toy-Directed Behaviours | | |
| **Behaviour** | **Behaviour Description** | |
| Moving Towards Toy | Distance between dog and toy (or toy basket) decreases, as dog moves with their head oriented towards the toy (or toy basket). It should be marked from the moment the dog lifts their paw to make the movement towards the toy / toy basket. | |
| Looking at Toy | Dog faces the toy (or toy basket) and looks at it for >1 second. | |
| Picking up Toy | Dog picks up toy (or toy basket) with their muzzle. | |
| Interacting with Toy | Dog intentionally physically interacts with a toy (or toy basket): namely by scratching at, pawing at, sniffing it, or nudging it. Sniffing involves a dog moving its nostrils while holding their muzzle still and in very close proximity (< 5cm) to the object. | |
| Food-Directed Behaviours | | |
| **Behaviour** | **Behaviour Description** | |
| Moving Towards Bowl | Distance between dog and food bowl decreases, as dog moves with their head oriented towards the food bowl. It should be marked from the moment the dog lifts their paw to make the movement towards the food / food bowl. | |
| Looking at Bowl | Dog faces the food bowl and looks at it for >1 second. | |
| Sitting/Lying by Bowl | Dog changes their position to sitting by or laying down by the food bowl. | |
| Interacting with Bowl | Dog intentionally physically interacts with empty food bowl: namely by scratching at, pawing at, sniffing it, or nudging it. Sniffing involves a dog moving its nostrils while holding their muzzle still and in very close proximity (< 5cm) to the object. | |

was cut so that it began following the moment that E1 stood upright after pressing a button and ended 60 seconds later.

Blind coders were naïve to experimental design, experimental hypotheses, and conditions. They were initially trained on a set of videos which included trials of the study piloted with non-eligible dogs and non-study videos crowdsourced online of non-subjects. Coders annotated videos following a pre-determined ethogram (Table 3). Interrater agreement was substantial throughout training (Kappa coefficient of 0.81, with a confidence interval of [0.76; 0.86] for 100% of training set) and remained satisfactory throughout the coding of the full dataset (Kappa coefficient of 0.70, with a confidence interval of [0.64; 0.76] for a randomly selected subset comprising 10% of all data).

To determine whether dogs responded as expected to E1's button presses (in Experiment 1) and the owners' button presses (in Experiment 2), we investigated whether their behavioural responses to button presses were consistent with individual buttons' words. For example,

upon observing a button press for "OUTSIDE", a dog should be more likely to move toward the door than upon observing a press for "FOOD".

The data was analysed using Bayesian linear mixed-effects models implemented in *brms* [21] in R [22]. Specifically, we used a Bernoulli model which requires the dependent variable to be binary. In our case, we ran three separate Bernoulli models, one for each type of behaviour (food-directed behaviours, outside-directed behaviours, and play-directed behaviours). Our reasoning was that if dogs have created associations between these words and their meanings, then they should show the appropriate behaviour associated with that button in the condition.

The dependent variable for each model was whether, on any given trial, the dog showed the target behaviour. For example, in the model for the FOOD condition, the dependent variable was 1 for a given trial if the dog showed a food-related behaviour, and 0 if not (where "trial" here refers to a coded behaviour, so each food-related behaviour would be coded 1 in the FOOD model, and each non-food behaviour would be coded 0). A complete model description can be found in the analysis script (which is provided in full at https://github.com/znhoughton/comprehension).

The independent variables were the three conditions (Food Condition, Outside Condition, or Play Condition), Experiment (IPS or CSS), and the interaction between the two, with maximal random effects [23]. For each model, we used weakly informative priors. The model syntax for each model is included below:

Model 1: Play behaviours ~ Condition*Experiment + (1 + Condition|Subject)

Model 2: Outside behaviours ~ Condition*Experiment + (1 + Condition|Subject)

Model 3: Food behaviours ~ Condition*Experiment + (1 + Condition|Subject)

Next, using only data from Experiment 2 (CSS), we tested whether dogs behave comparably in response to a spoken word and a word produced by a button press. To do so, we once again used three Bayesian logistic regression models implemented in brms in R, the same dependent variable as above, but with the fixed-effects of condition and mode (i.e., spoken or pressed; and their interaction), random slopes for condition by subject. This is described in the following three equations:

Model 4: Food behaviours ~ Condition*Mode + (1 + Condition*Mode|Subject)

Model 5: Outside behaviours ~ Condition*Mode + (1 + Condition*Mode|Subject)

Model 6: Play behaviours ~ Condition*Mode + (1 + Condition*Mode|Subject)

Finally, given recent work to suggest head-tilting is a lateralized behaviour which occurs when dogs recognise and process known words [18], we planned to compare the number of head tilts (to both sides first, then separately for the right and left sides) dogs made in the nonce ("Daxing") condition compared to the three meaningful known-word conditions (Outside, Play, and Food conditions) in both the researcher-led experiment and the citizen science component of this study. In order to assess this, we set out to use a Bayesian linear mixed-effects model with a Poisson distribution implemented in brms in R, with number of head tilts as the dependent variable and condition and context (known word vs. nonce word) as well as their interaction as fixed-effects and random intercepts for subject and random slopes for condition by subject. Our model specification is listed in the following equation: Number of head tilts ~ Condition * Experiment + (1|Subject) + (Condition|Subject).

All analyses were pre-registered in the Open Science Framework (https://osf.io/tskcq).

**Table 4. Output for Model 1: Play behaviours ~ Condition\*Experiment + (1 + Condition|Subject).** In the Play Condition, the odds of dogs displaying play-directed behaviours was approximately seven times greater than average across the three meaningful-word conditions.

| | Estimate | Est.Error | Q2.5 | Q97.5 |
|---|---|---|---|---|
| **Intercept** | -1.771 | 0.953 | -3.746 | -0.021 |
| **Play Condition** | 1.968 | 0.943 | 0.314 | 4.026 |
| **Food Condition** | 0.099 | 0.889 | -1.695 | 1.828 |
| **Outside Condition** | -2.067 | 1.355 | -5.104 | 0.270 |
| **Experiment CSS** | 0.555 | 0.816 | -0.994 | 2.277 |
| **Play:CSS** | -0.758 | 0.848 | -2.591 | 0.744 |
| **Food:CSS** | -0.157 | 0.794 | -1.740 | 1.381 |

## Results

### Question 1: Can dogs recognise and respond appropriate to the words recorded on their soundboards?

Across both experiments, dogs exhibited approximately seven times more play-directed behaviours in the Play Condition (Table 4), and approximately seven times more outside-directed behaviours in the Outside Condition (Table 5), suggesting that they recognised and responded appropriately to these two words. We found no conclusive evidence to suggest that dogs exhibited food-directed behaviours in the Food Condition compared to the other two conditions (Table 6). Note that, since all models used sum coding, the intercept in all tables represents the grand mean, and the coefficient values represent the distance, in log-odds, between the effect and the intercept. Dogs' behaviours across the three familiar-word conditions of both experiments are shown in Fig 2.

### Question 2: Do dogs respond equivalently to words produced by their owner compared to an unfamiliar person?

We found no effect of button presser identity (owner or unfamiliar person) on dog's behaviours for any of the three conditions (see confidence intervals overlapping zero in Tables 5–7 for IPS and CSS comparisons). This suggests that dogs respond appropriately to button presses even in the absence of other contextual cues or owner-produced cues.

### Question 3: Can citizen science studies and in-person studies of soundboard-trained dogs produce comparable results?

We found no difference in dogs' behaviours in response to owner-produced and experimenter-produced button presses (see Tables 4–6), and therefore the results for both

**Table 5. Output for Model 2: Outside behaviours ~ Condition\*Experiment + (1 + Condition|Subject).** In the Outside Condition, the odds of dogs displaying outside-directed behaviours was approximately seven times greater than average across the three meaningful-word conditions.

| | Estimate | Est.Error | Q2.5 | Q97.5 |
|---|---|---|---|---|
| **Intercept** | -1.725 | 0.875 | -3.572 | -0.061 |
| **Outside Condition** | 1.999 | 0.852 | 0.423 | 3.806 |
| **Food Condition** | -1.513 | 0.839 | -3.326 | 0.029 |
| **Play Condition** | -0.485 | 0.812 | -2.145 | 1.039 |
| **Experiment CSS** | -1.906 | 0.780 | -3.583 | -0.503 |
| **Outside:CSS** | -0.030 | 0.865 | -1.885 | 1.563 |
| **Food:CSS** | -0.220 | 0.790 | -1.811 | 1.290 |

**Table 6. Output for Model 3: Food behaviours ~ Condition*Experiment + (1 + Condition|Subject).** Dogs were not more likely to perform food-directed behaviours in the Food Condition compared to Play or Outside Conditions.

|  | Estimate | Est.Error | Q2.5 | Q97.5 |
|---|---|---|---|---|
| Intercept | -4.110 | 0.823 | -5.943 | -2.715 |
| Food Condition | 1.279 | 0.828 | -0.260 | 3.008 |
| Outside Condition | -0.040 | 0.977 | -2.134 | 1.778 |
| Play Condition | -1.239 | 1.122 | -3.698 | 0.703 |
| Experiment CSS | -0.199 | 0.557 | -1.311 | 0.880 |
| Food:CSS | 0.050 | 0.609 | -1.132 | 1.296 |
| Outside:CSS | -0.303 | 0.746 | -1.883 | 1.082 |

experiments were comparable. In the present investigation, dogs behaved similarly in response to button presses in the in-person study (IPS) and the citizen science study (CSS), suggesting that both experiments produced equivalent results.

## Question 4: Do soundboard-trained dogs attend to speech?

Within Experiment 2 (IPS), dogs' responses to spoken and pressed words across the three conditions were comparable for food-related behaviours (Table 7), outside-related behaviours (Table 8), and play-related behaviours (Table 9). Overall, these results suggest that the two modes of word-production led to equivalent behavioural responses by subjects.

**Question 5: Do soundboard-trained dogs tilt their heads in response to familiar words?** Ultimately, our dataset contained too few head-tilting observations (9 instances) for this analysis to be completed, so we cannot ascertain whether head-tilting was more common in meaningful known-word conditions (Play, Food, and Outside Conditions) compared to the nonce word condition (Daxing Condition).

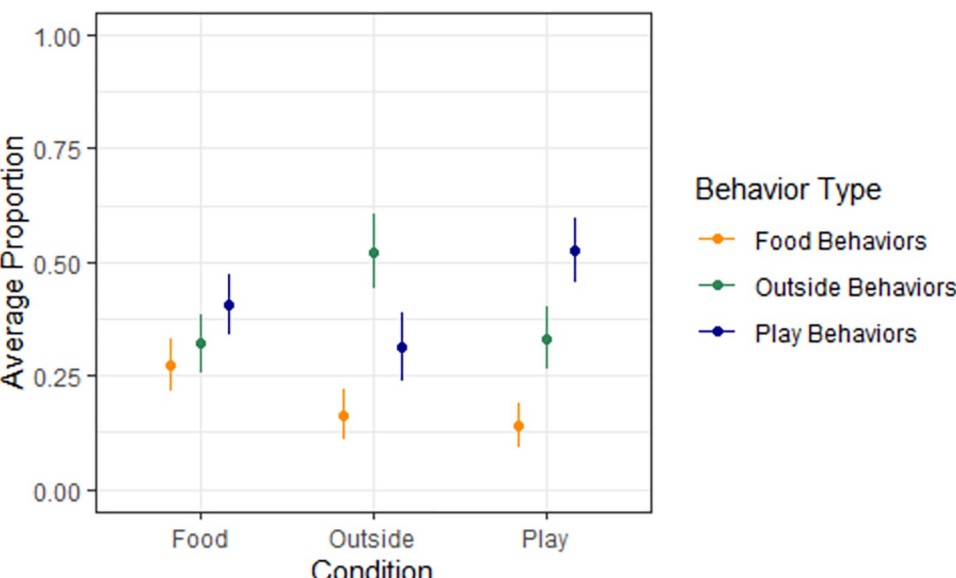

**Fig 2. The plot of the average proportion of each behaviour type in each condition.** Proportion here is the number of behaviours of a specific type (e.g., outside behaviours) in a given condition (e.g., play condition) divided by the total number of behaviours in that condition. The x-axis is Condition. The y-axis is the average proportion of each behaviour type across participants. Each of the points represents the proportion of behaviours of the corresponding type of behaviour in the respective condition. The error bars represent standard errors.

**Table 7. Output for Model 4: Food behaviours ~ Condition*Mode + (1 + Condition*Mode|Subject).** "Play:Press", "Food:Press" and "Outside:Press" correspond to the interactions between each condition and mode. For example, the effect of "Play:Press" is the extent to which the effect of "Play" differs when the dog hears a recording from a button press as opposed to a spoken word.

| | Estimate | Est.Error | Q2.5 | Q97.5 |
|---|---|---|---|---|
| Intercept | -5.749 | 1.373 | -8.740 | -3.494 |
| Play | 1.680 | 1.299 | -0.855 | 4.388 |
| Food | 0.315 | 1.262 | -2.269 | 2.759 |
| Press | 0.261 | 0.775 | -1.137 | 1.915 |
| Play:Press | -1.155 | 1.059 | -3.377 | 0.846 |
| Food:Press | -0.417 | 1.136 | -2.687 | 1.890 |

## Discussion

Our study suggests that dogs were more likely to perform play-related behaviours after an experimenter or their owner produced a play-related word, and were more likely to exhibit outside-related behaviours in response to an experimenter or their owner producing an outside-related word. This demonstrates that dogs are, at the very least, capable of learning an association between these words or buttons and their outcomes in the world. On the other hand, dogs did not produce preferentially food-related behaviours in response to the relevant words. It is possible that this may have occurred because several of the dogs taking part in the study may have been satiated before the start of test trials, or because dogs did not expect that they would be served a meal outside of their usual feeding times. Our in-person study was typically conducted during work hours, which do not typically overlap with adult dogs' mealtimes in the early mornings and/or evenings.

Given that dogs responded equivalently to their owners' button presses and an unfamiliar person's button presses, our results also demonstrate that dogs attend to and respond to the buttons or words themselves, rather than behaving solely based on unrelated unintentional cues provided by their owners (e.g., as would be expected from a Clever Hans effect). In both experiments, even when all word-related contextual cues were removed from the human's interaction with the AIC device, dogs still responded with contextually appropriate behaviours. For example, if dogs typically observe their owners picking up a toy before they press a button for a play-related word, any play-related behaviours the dogs exhibit in turn might occur in response to the sight of the toy, the owner's button press, or a combination of both events. Since our citizen science experiment specifically required that owners press buttons without performing any other actions, the fact that dogs still displayed contextually appropriate responses in the absence of other cues demonstrates that they attended specifically to the

**Table 8. Output for Model 5: Outside behaviours ~ Condition*Mode + (1 + Condition*Mode|Subject).** "Play:Press", "Food:Press" and "Outside:Press" correspond to the interactions between each condition and mode. For example, the effect of "Play:Press" is the extent to which the effect of "Play" differs when the dog hears a recording from a button press as opposed to a spoken word.

| | Estimate | Est.Error | Q2.5 | Q97.5 |
|---|---|---|---|---|
| Intercept | -5.978 | 1.941 | -10.253 | -2.603 |
| Outside | 3.251 | 1.436 | 0.580 | 6.307 |
| Food | -2.180 | 1.539 | -5.391 | 0.741 |
| Press | 0.245 | 1.178 | -2.034 | 2.618 |
| Outside:Press | -0.328 | 1.495 | -3.386 | 2.599 |
| Food:Press | 2.842 | 1.563 | 0.197 | 6.344 |

**Table 9. Output for Model 6: Play behaviours ~ Condition\*Mode + (1 + Condition\*Mode|Subject).** "Play:Press", "Food:Press" and "Outside:Press" correspond to the interactions between each condition and mode. For example, the effect of "Play:Press" is the extent to which the effect of "Play" differs when the dog hears a recording from a button press as opposed to a spoken word.

| | Estimate | Est.Error | Q2.5 | Q97.5 |
|---|---|---|---|---|
| Intercept | -1.027 | 1.317 | -3.754 | 1.482 |
| Food | 2.154 | 1.170 | 0.085 | 4.699 |
| Outside | -0.188 | 0.949 | -2.139 | 1.710 |
| Press | 1.275 | 0.711 | -0.044 | 2.755 |
| Food:Press | -0.437 | 0.775 | -2.020 | 1.099 |
| Outside:Press | -0.784 | 0.737 | -2.263 | 0.625 |

words recorded onto their buttons. Additionally, dogs responded appropriately to an unfamiliar human making button presses, even though that person was unaware of which button they were pressing.

Both word sounds, and the location of their respective buttons on dogs' soundboards, are highly correlated with their effects in the world. For example, when owners press the button OUTSIDE, dogs might equally attend to the location of that button on their AIC device, or the sound (word) produced by the button press. Our citizen science study addressed this question by comparing dogs' responses to their owners' speech and button presses. We found no differences in dogs' responses to either mode of word production, suggesting that most dogs do not associate button location *alone* with the outcomes of button presses. If they did, they should perform considerably more context-appropriate behaviours in response to the relevant button presses than to owners' spoken words. We note that, although this does not necessarily preclude that dogs may have formed two separate associations–one for the button location and another for the speech sound of the word–, it nevertheless demonstrates that most dogs can and do attend to and appropriately respond to the auditory properties of words.

Crucially, we found no differences in dogs' behaviours across the two experiments, suggesting that owners' and researchers' conduct of the methods was sufficiently equivalent to yield comparable results. This suggests that results from the citizen science version of the study were comparable to those performed by researchers during visits to owners' homes. This is important because soundboard-trained dogs are spread all over the world and we do not yet know

**Table 10. Summary of the conclusions drawn for each of the five questions addressed in the present study.**

| Research Question | Results of Study |
|---|---|
| 1. Can dogs recognise and respond appropriate to the words recorded on their soundboards? | Yes; dogs demonstrated context-appropriate responses to two of the three word categories tested, namely "play" and "outside". However, we found no evidence for context-appropriate responses to "food"-related words. |
| 2. Do dogs respond equivalently to words produced by their owner compared to an unfamiliar person? | Yes; we did not find any differences in dogs' responses to button presses produced by their owners compared to an unfamiliar person. |
| 3. Can citizen science studies and in-person studies of soundboard-trained dogs produce equivalent results? | Yes; in this case, the citizen science study yielded comparable results to the in-person study carried out by researchers. |
| 4. Do soundboard-trained dogs attend to speech? | Yes; dogs responded equivalently to their owners pressing words on their soundboards compared to their owners speaking these same words. |
| 5. Do soundboard-trained dogs tilt their heads in response to familiar words? | Inconclusive; we did not observe enough head tilting in our dataset to investigate this question. |

how moving their soundboards outside of their home environment might affect their soundboard use [2]. Our findings offer a promising outlook for future citizen science studies with this population of owners and their dogs: remotely conducted citizen science studies could be a critical tool for studying this large and geographically widespread population and maintaining long-term engagement of pet owners with this research program. However, we advise that other future studies should again validate citizen science methods by comparing them against researcher-led in-person experiments, therefore replicating our findings, before scientists can rely more heavily on citizen science for this and similar studies. Additionally, we highlight that all procedures and analyses of this study were pre-registered in advance of data collection, and that future studies extending on this work would also benefit from pre-registration. Pre-registration offers an important tool for ensuring transparency and reproducibility of research, and its use is critical to large scale exploratory studies and citizen science studies.

In sum, our findings provide the first evidence of button word comprehension by owner-trained soundboard-using dogs, and demonstrate that dogs' contextually appropriate responses to button presses were comparable regardless of the identity of the person using the soundboard, and the absence of other environmental cues related to that word. Our findings also suggest that dogs attend to the sounds recorded onto their buttons, given that they responded equivalently to words when they were produced by button presses and when they were spoken by their owners. A summary of our findings is tabulated in Table 10 below.

In our study, dogs responded to spoken or pressed food-related and outside-related words with contextually appropriate responses slightly more often than expected by chance. The accuracy of dog's responses in our study is comparable to dogs' accuracy in responding to human pointing [24]. However, dogs are capable of much greater accuracy when purposely trained to respond to stimuli, as is the case for dogs trained to detect wildlife [25], individual people [26], and chemical substances [27, 28] through scent. This discrepancy in performance could be due to the nature of the stimuli: perhaps dogs find it more difficult to form associations between words and their respective outcomes compared to scents and a consistent reward. Alternately, communicative contexts may produce less predictable responses due to the weaker correlation between the perception of relevant stimuli (a pointed finger, or a word) and their outcomes, therefore leading to less strongly conditioned responses to the stimuli. While scent detection dogs undergoing training will likely experience reinforcement almost every single time the target stimulus is present, a pet dog may not receive any reinforcement on a great number of occasions when they hear familiar words or observe human pointing. Relatedly, scent detection dogs' accuracy is much higher in contexts where target odours are present at very high rates, with implications for their performance in real-world scenarios [29]. Therefore, it is also possible that task-oriented trained contexts, such as scent detection, are much more motivating to dogs, and therefore more likely to trigger consistent responses, than day-to-day contexts involving more variable reinforcement. Further, measures of performance in working dogs such as scent detection dogs are typically based on a small subset of the population that passes stringent standards of training [30], and typically involves animals that are selectively bred for the purpose of scent detection and further selected based on temperament or cognitive traits [30–32], whereas the soundboard-trained population comprises owner-trained pet dogs, whose temperaments and cognitive traits are likely to vary considerably.

Having established that soundboard-trained dogs can and do attend to and comprehend words, future work should also disambiguate the extent to which spatial information about button positions, or other potential cues for button identity, might aid dogs' ability to use AIC devices. Additionally, more research is needed investigating soundboard-trained dogs' responses to a wider range of words, particularly in comparison to a population of non-soundboard-trained pet dogs. Although owners anecdotally report that owner-trained pet dogs

spontaneously acquire comprehension of large spoken vocabularies [9], there are no fully controlled experiments investigating whether dogs exhibit contextually appropriate spontaneous responses to familiar words in the absence of other contextual cues, as in the present study. This is crucial because, although owners may, for example, report that their dogs respond appropriately to a food-related word, this word is typically presented alongside a myriad of other confounding cues, such as the time of day when the dog's meals are served, the presence of a bowl, or the behaviours their owner might perform before serving their dog's dinner. Finally, our ongoing work is investigating dogs' word production [2]. In order to determine whether dogs' performance at word comprehension is reflected in their button pressing, carefully controlled future studies must investigate whether dogs can spontaneously produce contextually appropriate button presses in experimentally induced situations (as in [4]). Not only would such a study be helpful in understanding the depth of soundboard-trained dogs' word comprehension, but it would also establish the extent to which AIC devices can be used for two-way interspecies communication, involving both presses made by the owner for their dog, and by the dog for their owner.

## Acknowledgments

We are grateful to all our dog participants and their owners for contributing their time and data to our research project.

## Author Contributions

**Conceptualization:** Amalia P. M. Bastos, Ashley Evenson, Gabriella E. Smith, Jack Terwilliger, Federico Rossano.

**Data curation:** Amalia P. M. Bastos.

**Formal analysis:** Zachary N. Houghton, Lucas Naranjo.

**Investigation:** Amalia P. M. Bastos, Ashley Evenson, Patrick M. Wood, Alexandria Cairo-Evans, Lisa Korpos, Sarita Raghunath, Cassandra Paul, Hairou Hou.

**Methodology:** Amalia P. M. Bastos, Ashley Evenson, Patrick M. Wood, Gabriella E. Smith, Federico Rossano.

**Project administration:** Amalia P. M. Bastos, Patrick M. Wood, Federico Rossano.

**Resources:** Patrick M. Wood, Federico Rossano.

**Software:** Patrick M. Wood.

**Supervision:** Federico Rossano.

**Visualization:** Zachary N. Houghton.

**Writing – original draft:** Amalia P. M. Bastos.

**Writing – review & editing:** Amalia P. M. Bastos, Zachary N. Houghton, Lucas Naranjo, Gabriella E. Smith, Jack Terwilliger, Federico Rossano.

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
