## [Decision Letter · Decision Letter 0]

30 May 2024

PONE-D-24-15704How do soundboard-using dogs respond to human button presses? An investigation into label comprehensionPLOS ONE

Dear Dr. Bastos,

Thank you for submitting your manuscript to PLOS ONE. After careful consideration, we feel that it has merit but does not fully meet PLOS ONE’s publication criteria as it currently stands. Therefore, we invite you to submit a revised version of the manuscript that addresses the points raised during the review process.  Please carefully address the reviewer's concerns regarding the presentation of your results and conclusions of your study in your revision.  This includes potentially altering the title of your manuscript as you revise based on the thoughtful critique provided by the reviewer. 

We look forward to receiving your revised manuscript.

Kind regards,

Brenton G. Cooper, Ph.D.

Academic Editor

PLOS ONE

“A.P.M.B. is supported by the Johns Hopkins Provost’s Postdoctoral Fellowship Program.”

“A.P.M.B., P.M.W., Z.N.H., J. T., G.E.S., & L.K. have previously consulted for FluentPet, Inc., a company that produces AIC devices for pets. A.E. & L.N. are employees of FluentPet, Inc.”

6. Please remove your figures from within your manuscript file, leaving only the individual TIFF/EPS image files, uploaded separately. These will be automatically included in the reviewers’ PDF.

Reviewers' comments:

Reviewer's Responses to Questions

**Comments to the Author**

1. Is the manuscript technically sound, and do the data support the conclusions?

Reviewer #1: Partly

2. Has the statistical analysis been performed appropriately and rigorously? 

Reviewer #1: Yes

3. Have the authors made all data underlying the findings in their manuscript fully available?

Reviewer #1: Yes

4. Is the manuscript presented in an intelligible fashion and written in standard English?

Reviewer #1: Yes

5. Review Comments to the Author

Reviewer #1: Bastos et al.

How do soundboard-using dogs respond to human button presses?...

May 14, 2024

At its heart this is a very simple paper, but, because of the context of this work, it is likely to excite a lot of attention. Much of it misguided. Consequently, a great deal of care needs to be taken in its presentation. A lot of care has already been taken. I’m going to suggest some additional care.

In essence, Bastos et al. tested pet dogs for their ability to respond appropriately to three words: “Out,” “play,” and “food.” (The exact word used depended on the dog’s prior training). “Respond appropriately” means here, move towards the room exit in response to “out,” move towards toys in response to “play,” and move towards food in response to “food.” The authors found above chance tendencies to move towards the exit and toys in response to “out,” and “play” respectively. There was no tendency to move towards food in response to “food.” These tendencies to respond appropriately to “out” and “play” were not massive effects. On average the dogs showed about 50% appropriate behaviors in response to the relevant word. So a detectable tendency but far from the levels expected, say, of explosive detection dogs.

The study stands out for extreme care in design and experimental controls. It was preregistered, every imaginable attempt was made to blind the experimenters and video coders to what was going on. So the risks of Clever-Hans-type unintentional cueing are very small. The data analysis also appears careful and fully appropriate.

Thus far this might seem like a very minor study. What will surely lead to it attracting a great deal of attention is the fact that the dogs that were tested had all been trained (by their owners) on the button systems that are currently very popular in social media and elsewhere. These are buttons programmed so that, when pressed, they say a word recorded by the dog’s owner. The widespread interest in these buttons stems from the fact that people train their dogs to press them, however, in this study, the dogs were not given an opportunity to press the buttons. All button presses were carried out by humans. There were two sets of conditions: one where the owner or experimenter said the relevant word; the other where the owner or experimenter generated the word by pressing the button that activated a prior recording of the owner saying the word. No significant differences were found between these conditions. Furthermore, there is no basis to believe that the fact that the dogs had all been trained to press these buttons themselves has anything to do with their performance. It wouldn’t surprise me if most pet dogs respond 50% appropriately to someone saying “out” or “play.” In fact, I’m rather surprised that the dogs in this study didn’t respond correctly to a word standing for “food.” My dog does and we haven’t had to train her explicitly to respond to “din dins!”

There were also a further two sets of conditions, one in which the dogs were tested by experimenters, and one where the dogs were tested by their owners. No difference was found between these conditions.

I think this is an interesting little paper, but, as I started out by stating, its presentation and conclusions need to be carefully expressed to avoid the possibility of misunderstandings.

First, I think the fact that these dogs had been trained to press “talking” buttons should be downplayed. In this study the dogs were not pressing the buttons. There are no comparative data to indicate that the prior button-press-training has anything to do with the performance observed, and the generally low (albeit statistically significantly above chance) levels of performance provide no grounds for believing that the prior training had had any positive effect.

Figure 2 should be redrawn to cover the whole range over which values were free to vary (0 – 100%). The present truncated formatting gives a visual impression of a more powerful effect than was actually observed.

I wonder about including “soundboard-using” in the title. As noted above, there is no evidence that having been trained on the soundboard impacted the dogs’ performance.

Tables 8-10 lack full explanations. What are “Play:Press” and “Food:Press” etc.?

I’m not sure why, in the Discussion, the authors stated that they possibly failed to detect a respond to the word “food” because food-related behaviors occurred at a high rate in all conditions. Figure 2 indicates the opposite; Food related behaviors occur at the lowest rate of any analyzed behaviors in all three conditions.

I recognize that this isn’t an error, but I think it would be simpler to refer to the stimuli as ‘words” rather than “labels.” That will be easier for more readers to understand.

I would like to see consideration in the Discussion of other contexts in which dogs respond to stimuli. This might include scenarios like sniffer dogs, which are trained explicitly and to far higher criteria than were observed here, and possibly other everyday examples of dogs developing appropriate responses in daily interaction with humans, such as following pointing gestures.

Overall a worthy piece of work that ought to be published after some revision.

6. PLOS authors have the option to publish the peer review history of their article (what does this mean?). If published, this will include your full peer review and any attached files.

Reviewer #1: No

---

## [Author Response · Author response to Decision Letter 0]

10 Jun 2024

Reviewer Comment: At its heart this is a very simple paper, but, because of the context of this work, it is likely to excite a lot of attention. Much of it misguided. Consequently, a great deal of care needs to be taken in its presentation. A lot of care has already been taken.

I’m going to suggest some additional care. In essence, Bastos et al. tested pet dogs for their ability to respond appropriately to three words: “Out,” “play,” and “food.” (The exact word used depended on the dog’s prior training). “Respond appropriately” means here, move towards the room exit in response to “out,” move towards toys in response to “play,” and move towards food in response to “food.” The authors found above chance tendencies to move towards the exit and toys in response to “out,” and “play” respectively. There was no tendency to move towards food in response to “food.” These tendencies to respond appropriately to “out” and “play” were not massive effects. On average the dogs showed about 50% appropriate behaviors in response to the relevant word. So a detectable tendency but far from the levels expected, say, of explosive detection dogs.

The study stands out for extreme care in design and experimental controls. It was preregistered, every imaginable attempt was made to blind the experimenters and video coders to what was going on. So the risks of Clever-Hans-type unintentional cueing are very small. The data analysis also appears careful and fully appropriate.

Author Response: We thank the reviewer for their positive comments.

Reviewer Comment: Thus far this might seem like a very minor study. What will surely lead to it attracting a great deal of attention is the fact that the dogs that were tested had all been trained (by their owners) on the button systems that are currently very popular in social media and elsewhere. These are buttons programmed so that, when pressed, they say a word recorded by the dog’s owner. The widespread interest in these buttons stems from the fact that people train their dogs to press them, however, in this study, the dogs were not given an opportunity to press the buttons. All button presses were carried out by humans. There were two sets of conditions: one where the owner or experimenter said the relevant word; the other where the owner or experimenter generated the word by pressing the button that activated a prior recording of the owner saying the word. No significant differences were found between these conditions. Furthermore, there is no basis to believe that the fact that the dogs had all been trained to press these buttons themselves has anything to do with their performance.

Author Response: The reviewer is correct in that this is a study of comprehension rather than production: at no point did the dogs have to press any buttons. We agree that some non-soundboard-trained dogs also likely recognise certain words (e.g., “food”), but we cannot at this point generalise this finding to all pet dogs. Soundboard-trained dogs experience human speech directed at them very regularly. This may not be true of many pet dogs (e.g., particularly those living outside the home in a yard, or in cultures where pets may not be typically spoken to).

Reviewer Comment: It wouldn’t surprise me if most pet dogs respond 50% appropriately to someone saying “out” or “play.” In fact, I’m rather surprised that the dogs in this study didn’t respond correctly to a word standing for “food.” My dog does and we haven’t had to train her explicitly to respond to “din dins!”

Author Response: We thank the reviewer for this excellent example. We have used it to improve our discussion, raising the points the reviewer made about comparing soundboard-trained dogs and non-soundboard-trained dogs with regards to their word comprehension: “Additionally, more research is needed investigating soundboard-trained dogs’ responses to a wider range of words, particularly in comparison to a population of non-soundboard-trained pet dogs. Although owners anecdotally report that owner-trained pet dogs spontaneously acquire comprehension of large spoken vocabularies [9], there are no fully controlled experiments investigating whether dogs exhibit contextually appropriate spontaneous responses to familiar words in the absence of other contextual cues, as in the present study. This is crucial because, although owners may, for example, report that their dogs respond appropriately to a food-related word, this word is typically presented alongside a myriad of other confounding cues, such as the time of day when the dog’s meals are served, the presence of a bowl, or the behaviours their owner might perform before serving their dog’s dinner. Finally, our ongoing work is investigating dogs’ word production [2]. In order to determine whether dogs’ performance at word comprehension is reflected in their button pressing, carefully controlled future studies must investigate whether dogs can spontaneously produce contextually appropriate button presses in experimentally induced situations (as in [4]).” (L520-534)

Reviewer Comment: There were also a further two sets of conditions, one in which the dogs were tested by experimenters, and one where the dogs were tested by their owners. No difference was found between these conditions.

I think this is an interesting little paper, but, as I started out by stating, its presentation and conclusions need to be carefully expressed to avoid the possibility of misunderstandings.

Author Response: We agree with the reviewer and thank them for their caution. We also believe this work requires careful presentation.

Reviewer Comment: First, I think the fact that these dogs had been trained to press “talking” buttons should be downplayed. In this study the dogs were not pressing the buttons. There are no comparative data to indicate that the prior button-press-training has anything to do with the performance observed, and the generally low (albeit statistically significantly above chance) levels of performance provide no grounds for believing that the prior training had had any positive effect.

Author Response: Although we agree that we cannot provide a direct comparison, it is very likely that owners who train their dogs to press buttons also direct a great deal of speech to their dogs – probably more so than the average pet owner. As such, these dogs may experience far more opportunities in which to learn the associations between words and their outcomes, compared to the average pet dog. We also cannot make any claims about how non-soundboard-trained pet dogs might react to a word – even if it is a familiar word – being produced by a button containing a voice recording. The training experienced by soundboard-using dogs at the very least habituates them to this context and enables them to respond to the voice recordings on buttons equivalently to speech.

Reviewer Comment: Figure 2 should be redrawn to cover the whole range over which values were free to vary (0 – 100%). The present truncated formatting gives a visual impression of a more powerful effect than was actually observed.

Author Response: We have now redrawn Figure 2 as per the reviewer’s comments.

Reviewer Comment: I wonder about including “soundboard-using” in the title. As noted above, there is no evidence that having been trained on the soundboard impacted the dogs’ performance.

Author Response: In line with the reviewer’s comment, we have changed “soundboard-using” to “soundboard-trained” in the title and at several points throughout the manuscript, as appropriate. Although there is no evidence that being trained on a soundboard increases a dog’s ability to learn the associations between words and their outcomes, it is quite likely that dogs whose owners train them to press buttons experience two considerably different sets of circumstances compared to the average pet dog: (1) they experience many more opportunities for word-to-outcome associative learning, because their owners regularly direct speech at them – and anecdotally we know from owner reports that this speech is simplified, with particular features such as exaggerated pronunciation and repetitions (this is a topic for future investigation); (2) they are additionally at the very least habituated to, if not directly associatively learning from, the voice recordings being produced by buttons. As such, it would be precipitous for us to generalise our findings to all pet dogs, regardless of training history.

Reviewer Comment: Tables 8-10 lack full explanations. What are “Play:Press” and “Food:Press” etc.?

Author Response: We have now added the following description to the three tables’ descriptive captions, to clarify these rows: ““Play:Press”, “Food:Press” and “Outside:Press” correspond to the interactions between each condition and mode. For example, the effect of “Play:Press” is the extent to which the effect of “Play” differs when the dog hears a recording from a button press as opposed to a spoken word.”

Reviewer Comment: I’m not sure why, in the Discussion, the authors stated that they possibly failed to detect a respond to the word “food” because food-related behaviors occurred at a high rate in all conditions. Figure 2 indicates the opposite; Food related behaviors occur at the lowest rate of any analyzed behaviors in all three conditions.

Author Response: The reviewer is correct. We apologise for this mistake and have rephrased our discussion accordingly: “It is possible that this may have occurred because several of the dogs taking part in the study may have been satiated before the start of test trials, or because dogs did not expect that they would be served a meal outside of their usual feeding times. Our in-person study was typically conducted during work hours, which do not typically overlap with adult dogs’ mealtimes in the early mornings and/or evenings.” (L443-447)

Reviewer Comment: I recognize that this isn’t an error, but I think it would be simpler to refer to the stimuli as ‘words” rather than “labels.” That will be easier for more readers to understand.

Author Response: We thank the reviewer for this suggestion. We have now rephrased “labels” as “words” throughout the manuscript.

Reviewer Comment: I would like to see consideration in the Discussion of other contexts in which dogs respond to stimuli. This might include scenarios like sniffer dogs, which are trained explicitly and to far higher criteria than were observed here, and possibly other everyday examples of dogs developing appropriate responses in daily interaction with humans, such as following pointing gestures.

Author Response: We thank the reviewer for this excellent suggestion. We have now included a paragraph on this topic in our discussion: “In our study, dogs responded to spoken or pressed food-related and outside-related words with contextually appropriate responses slightly more often than expected by chance. The accuracy of dog’s responses in our study is comparable to dogs’ accuracy in responding to human pointing [24]. However, dogs are capable of much greater accuracy when purposely trained to respond to stimuli, as is the case for dogs trained to detect wildlife [25], individual people [26], and chemical substances [27,28] through scent. This discrepancy in performance could be due to the nature of the stimuli: perhaps dogs find it more difficult to form associations between words and their respective outcomes compared to scents and a consistent reward. Alternately, communicative contexts may produce less predictable responses due to the weaker correlation between the perception of relevant stimuli (a pointed finger, or a word) and their outcomes, therefore leading to less strongly conditioned responses to the stimuli. While scent detection dogs undergoing training will likely experience reinforcement almost every single time the target stimulus is present, a pet dog may not receive any reinforcement on a great number of occasions when they hear familiar words or observe human pointing. Relatedly, scent detection dogs’ accuracy is much higher in contexts where target odours are present at very high rates, with implications for their performance in real-world scenarios [29]. Therefore, it is also possible that task-oriented trained contexts, such as scent detection, are much more motivating to dogs, and therefore more likely to trigger consistent responses, than day-to-day contexts involving more variable reinforcement. Further, measures of performance in working dogs such as scent detection dogs are typically based on a small subset of the population that passes stringent standards of training [30], and typically involves animals that are selectively bred for the purpose of scent detection and further selected based on temperament or cognitive traits [30–32], whereas the soundboard-trained population comprises owner-trained pet dogs, whose temperaments and cognitive traits are likely to vary considerably..” (L493-516)

Reviewer Comment: Overall a worthy piece of work that ought to be published after some revision.

Author Response: We again thank the reviewer for their positive comments.

---

## [Decision Letter · Decision Letter 1]

2 Jul 2024

How do soundboard-trained dogs respond to human button presses? An investigation into word comprehension

PONE-D-24-15704R1

Dear Dr. Bastos,

We’re pleased to inform you that your manuscript has been judged scientifically suitable for publication and will be formally accepted for publication once it meets all outstanding technical requirements.

Kind regards,

Brenton G. Cooper, Ph.D.

Academic Editor

PLOS ONE

Additional Editor Comments (optional):

Reviewers' comments:

Reviewer's Responses to Questions

**Comments to the Author**

1. If the authors have adequately addressed your comments raised in a previous round of review and you feel that this manuscript is now acceptable for publication, you may indicate that here to bypass the “Comments to the Author” section, enter your conflict of interest statement in the “Confidential to Editor” section, and submit your "Accept" recommendation.

Reviewer #1: All comments have been addressed

2. Is the manuscript technically sound, and do the data support the conclusions?

Reviewer #1: Yes

3. Has the statistical analysis been performed appropriately and rigorously? 

Reviewer #1: Yes

4. Have the authors made all data underlying the findings in their manuscript fully available?

Reviewer #1: Yes

5. Is the manuscript presented in an intelligible fashion and written in standard English?

Reviewer #1: Yes

6. Review Comments to the Author

Reviewer #1: My thanks to the authors for taking my comments on board. I think the revised version is an excellent paper and it will be very interesting to see the responses it engenders.

7. PLOS authors have the option to publish the peer review history of their article (what does this mean?). If published, this will include your full peer review and any attached files.

Reviewer #1: No

---

## [Editor Report · Acceptance letter]

19 Jul 2024

PONE-D-24-15704R1 

PLOS ONE

Dear Dr. Rossano, 

I'm pleased to inform you that your manuscript has been deemed suitable for publication in PLOS ONE. Congratulations! Your manuscript is now being handed over to our production team.

Kind regards, 

on behalf of

Dr. Brenton G. Cooper 

Academic Editor

PLOS ONE